# Potential of Cold-Water Agriculture (ColdAg) in Mexico: Challenges and Opportunities for Sustainable Food Production

Alejandro García-Huante [1], Fernando Carlos Gómez-Merino [1,2,*], Libia Iris Trejo-Téllez [3] and Amelia López-Herrera [1]

1 Department of Plant Physiology, College of Postgraduates in Agricultural Sciences Montecillo Campus, Carretera México-Texcoco km 36.5, Montecillo, Texcoco 56264, State of Mexico, Mexico; garcia.alejandro@colpos.mx (A.G.-H.); lopez.amelia@colpos.mx (A.L.-H.)

2 Colaborative Research Group at the Department of Sustainable Agrofood Innovation, College of Postgraduates in Agricultural Sciences Córdoba Campus, Carretera Córdoba-Veracruz km 348, Manuel León, Amatlán de los Reyes 94953, Veracruz, Mexico

3 Laboratory of Plant Nutrition, Department of Soil Science, College of Postgraduates in Agricultural Sciences Montecillo Campus, Carretera México-Texcoco km 36.5, Montecillo, Texcoco 56264, State of Mexico, Mexico; tlibia@colpos.mx

* Correspondence: fernandg@colpos.mx

**Abstract:** To guarantee sustainable development at a global level, humanity currently faces serious challenges related to a greater demand and better distribution of food to meet the needs of the growing population in environments affected by global climate change (GCC), and in limiting conditions with respect to natural, genetic, financial, and technological resources. Therefore, there is a dire need to implement technologies that can guarantee food security and sovereignty around the world, enabling sustainable development for all nations. Cold-Water Agriculture (ColdAg) is an available technology that offers an alternative to conventional food production in coastal areas and islands of the tropical and subtropical belts worldwide, making use of cold deep ocean water. Mexico has places with the optimal environmental conditions to adapt this technology, mainly in coastal areas and islands, which can contribute to ensuring access to food and drinking water. Initial tests have shown its viability, while production costs need to be adjusted to make this technology accessible to the poorest populations, so that it can contribute to their sustainable development and wealth. In this review article, we aimed to critically analyze some of the most salient literature on the ColdAg technology and outline the feasibility of this technology to be implemented and exploited in Mexico to contribute to alternative food production in a sustainable manner.

**Keywords:** OTEC; deep ocean water; chilled-soil agriculture; soil cooling; coastline; islands



## 1. Introduction

The increase in greenhouse gas emissions and its effect on global climate change (GCC) have led to the need for the agricultural sector to take various adaptation measures. Although GCC may display both positive and negative effects on diverse ecosystems around the world, Mexico's geographic characteristics make it a highly vulnerable country to the adverse impacts of climate change [1]. Among the negative effects of GCC are the more frequent occurrence of intense heat waves, droughts, and floods, proliferation of pests and diseases, or anomalies in soil temperature, which affect food availability and access, and increase the variability in prices of agricultural products [2]. The measures implemented so far to mitigate these effects have not been sufficient, and it is expected that the global average temperature will increase by 1.5 °C by the beginning of the year 2030 and 2 °C by 2040; that is, at a faster rate than the previous 2000 years, with potential devastating consequences for life on the planet [3]. Another study predicts that by mid-century (2040–2069) there will be an increase of 2.8 °C in maximum temperature and 2.2 °C

in minimum temperature in various parts of the world [2]. Those increases will cause plant stress and lead to widespread reductions in crop yields [3]. On the other hand, there will be some temperate climate crops that, due to an increase in temperature and a greater amount of atmospheric $CO_2$, could increase their yields [4]. Given this scenario, the sustainable production of agri-food crops is uncertain, and the search for alternatives that ensure access to safe and nutritious foods in sufficient quantities becomes necessary to maintain life and promote good health of the world's population [5].

Within the global food context, low crop productivity is associated with a reduction in timely water supply, rainfall availability, and erratic and intense rainfall patterns, as well as the contamination of much of the epicontinental water bodies [6–8]. Although the Green Revolution contributed to the increase in global agricultural production, that production paradigm is no longer valid. Hence, the development and implementation of other sustainable strategies that may grant sustained food production and guarantee food security in climate change scenarios remain as daunting tasks [9–11]. In less developed countries, the damage caused by GCC can threaten food security and household economies, especially because of a considerable reduction in crop yields [11,12]. Indeed, under mild-to-extreme warming climate scenarios without adaptation options, temperature increases are expected to decrease the relative yield of most crops by −6.2% to −18.3% in tropical and temperate regions [13,14].

On the other hand, the effects of climate change aggravate the problems of poverty and inequality [15,16], especially in developing countries and those with emerging economies, as is the case of Mexico. In these regions, much of the wealth is lost during extreme weather events, and in turn they have fewer resources to deal with the negative impacts of natural disasters [17]. This situation creates a spiral of adverse reactions in which marginalized populations are more likely to suffer the negative effects of extreme weather events and lose more of their wealth [18].

Consequently, public policies aimed at seeking mitigation and adaptation strategies to global climate change must take into account poverty and inequality indicators to achieve sustainable development objectives [19]. In Mexico, warming is higher than the global average; 2020 was the warmest year on record, being 1.6 °C above the historical average [20], generating greater pressure on the living conditions of the population, economic activities, and food production. In Mexico the biggest water-consuming group is the agriculture and livestock sector, which uses 76% of available water, with 54% use efficiency due to losses or leaks [21,22]. An additional problem is the overexploitation of aquifers, which generates water stress and pollution, especially in the north and northwest of the country. In the coastal regions of Mexico, access to drinking water and sewage services is insufficient and unbalanced. Another factor is the intrusion of seawater in coastal regions and reduction in water recharge (32 bodies of water present seawater intrusion, especially in the northern part of the country and the Yucatan Peninsula) and the contamination of bodies of water (lakes, lagoons, rivers, coastal lagoons, and beaches) [21].

The collateral effects of climate change have generated conditions of very low water availability across a wide region of central and northern Mexico [21]. Furthermore, factors such as population growth, regional economic development, and concentration of population in some urban areas, will exacerbate the problem, especially in basins located in the center, north, and northwest of the country [22,23].

Given the great heterogeneity in the human development indices of Mexico, it is necessary to design and implement public policies in accordance with the development contexts of each region of the country [24]. Studies on poverty and food security carried out within the country [25–27] are based on a partial and fragmented analysis in relation to the right to food security due to the concealment of information that hampers our ability to have a broad understanding of the problem, a precise vision of the issues of both physical and economic accessibility to food, and a well-grounded analysis of its availability and sustainability [18,28–30]. To eradicate hunger and poverty and improve the population's social well-being levels, it is necessary for countries to ensure food security, that is, that

families are guaranteed, always, physical, social, and economic access to sufficient, safe, and nutritious foods that meet their daily energy needs and food preferences for an active and healthy life [30,31].

Currently, Mexico has 46.8 million people in poverty, of which 37.7 million people are living in some degree of non-extreme poverty and 9.1 million are in extreme poverty [18], while 65% of the country's coastal regions continue to be centers of poverty [18,22]. In the case of food security, 23.4 million Mexicans still lack access to nutritious and quality food, despite the fact that important progress has been made, especially in rural regions [18,22,32].

Agriculture is considered a major driving force behind the development of civilizations in the world and has been the foundation for humanity's progress. However, it has generated negative impacts on the environment, and it is estimated that each year 4.2 million Mg of agricultural pollutants are dumped into the soil, water, and air. Unfortunately, agriculture also contributes about 10% of greenhouse gas emissions in the country [33]. Furthermore, agricultural activity is one of the main causes of biodiversity loss [34,35]. Due to its effects on land degradation, salinization of water and soil, overexploitation of aquifers, and the reduction in genetic diversity, agriculture threatens its own future. Therefore, there is an urgent need to implement sustainable production strategies that allow for reversing these negative effects [34].

Marine renewable energy may offer solutions for sustainable development, especially through Ocean Thermal Energy Conversion (OTEC) systems, which take advantage of the temperature differences (20 °C or greater) between the surface and deep water (~1000 m) of the ocean to generate different products, including electrical energy, air conditioning, and desalinated water, as well as other applications, such as aquaculture and cold agriculture (ColdAg) [36–38]. A typical OTEC system is depicted in Figure 1.

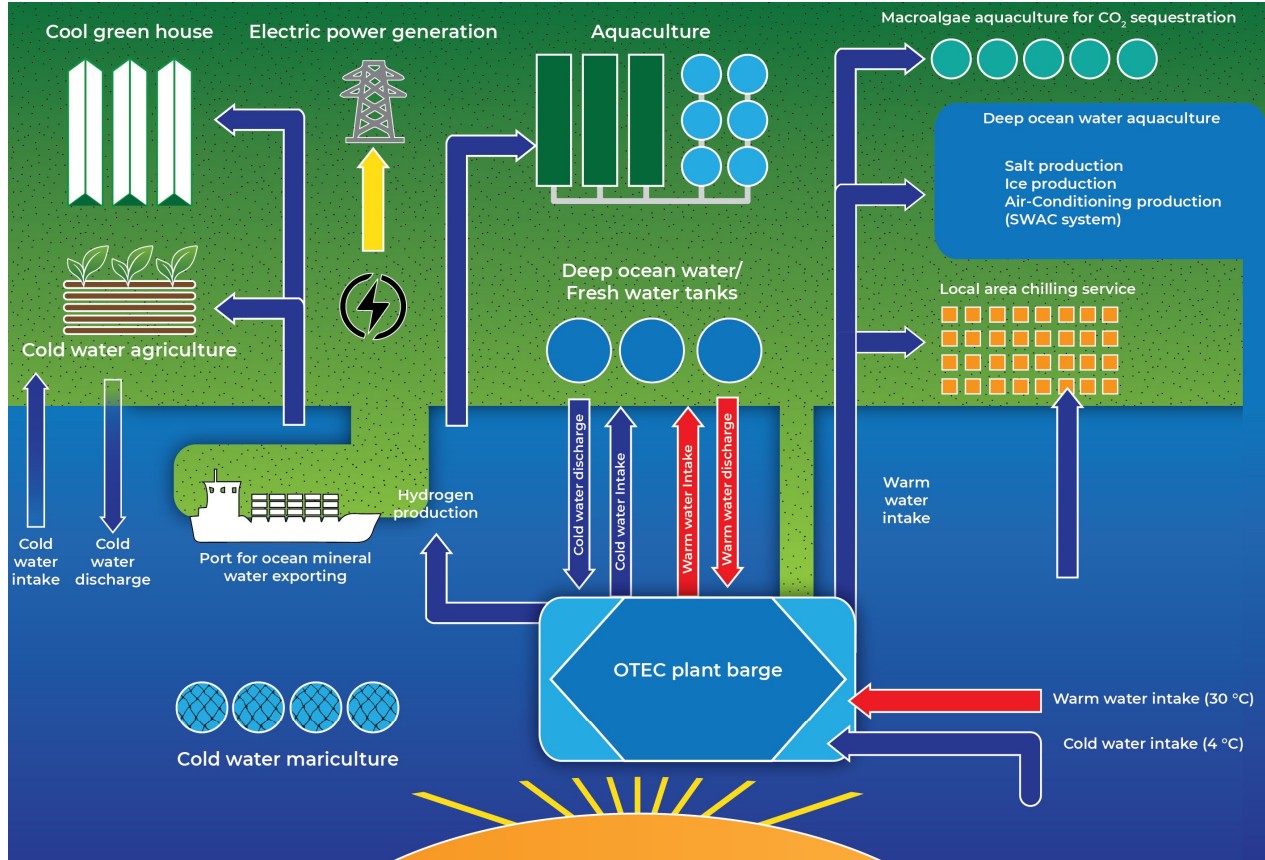

**Figure 1.** Open Cycle Ocean Thermal Energy Conversion (OTEC) system. The OTEC system may produce sustainable benefits in the areas electrical energy, drinking water provision, refrigeration/air conditioning, agriculture, aquaculture, and, even, mineral salt/cosmetic manufacture.

## 2. Cold-Water Agriculture (ColdAg) for Sustainable Food Production

In principle, ColdAg systems allow growing plants using depth-induced temperature difference between surface and deep seawater [36]. The soil is cooled by the temperature of cold deep water, and the temperature of the environment causes a thermal eustress in the plant with concomitant benefits in yield and quality of agri-food products [37–39]. Indeed, it is hypothesized that ColdAg would guarantee substantial increases in plant height, root and leaf lengths, and the sweetness and size of fruits [40].

In strawberry (*Fragaria* × *ananassa*), lettuce (*Lactuca sativa*), asparagus (*Asparagus officinalis*), and cabbage (*Brassica oleracea* var. *capitata*) qualitative yield analyses of the crops established on cold beds were in fact better than experimental controls [41]. The thermal gradients in the plants were directly proportional to the production or content of carbohydrates and proteins [39,40]. The system was patented and used to create a healthy and suitable soil environment for various plant species to grow and thrive in the harsh tropical coastal conditions [41].

Currently, research into ColdAg technology has advanced and work is now underway to optimize an efficient thermal energy transfer to agricultural systems. An upgrade over the pioneering ColdAg technology is improving thermal energy transfer and controls irrigation [42,43]. These systems efficiently control the biothermal energy potential of crops and the condensation irrigation rate using remote sensors and control systems that maximize the management of thermal energy resources [44], while a thermal energy system that captures fresh water from atmospheric vapor uses a cold thermal energy fluid through multiple linear heat exchangers, regulated by the temperature of the dew point. Their modular designs provide linear expansion to produce drinking and irrigation water in addition to crops [45].

## 3. Possibilities for Implementing ColdAg in Mexico

In some countries such as Japan and South Korea, OTEC systems have given rise to the development of both ColdAg and cold-water aquaculture. In Japan, research work was carried out on soil cooling for agriculture, including methods such as hydroponics and aquaponics, as well as cold-water aquaculture [46]. In Goseong, South Korea, the OTEC system has supported the development of companies dedicated to the production of food, salt, drinking water, fertilizers, and medicine. In the case of agriculture, research is being carried out where cold deep water is diluted between 50 and 70% to irrigate tomato (*Solanum lycopersicum*), chili (*Capsicum annuum*), and zucchini (*Cucurbita pepo*) crops under greenhouse conditions [36]. In Mexico, fishing and aquaculture represent important opportunities for sustainable development since the country has 11,592 km of coastline (73% corresponds to the Pacific Ocean and 27% to the Gulf of Mexico and the Caribbean Sea), with almost 3 million km$^2$ of exclusive economic zone, including 358 thousand km$^2$ of continental shelf [47] (Figure 2). This condition also represents a tremendous advantage to develop profitable OTEC systems, with both ColdAg and cold-water aquaculture as priority approaches.

Apart from the extensive coastline, Mexico's rivers and streams constitute a hydrographic network 633 thousand kilometers long, with a total of 2.9 million hectares of inland waters [47,48]. These characteristics provide great productive diversity and may strengthen both agriculture and aquaculture if sustainably and strategically managed [48,49]. Together with agriculture, fishing and aquaculture are matters of national security and are an essential part of the country's economic and social activities. Fishing production until 2021 in Mexico reached 1.92 million Mg of weight (fresh basis), with the Pacific coast providing 84%, while the coast of the Gulf of Mexico and the Caribbean Sea contributes with 16% of the total catch [49]. Aquaculture participates in Mexico's fish production with just over 18% of the total, with an average annual growth rate of 5.3% [50].

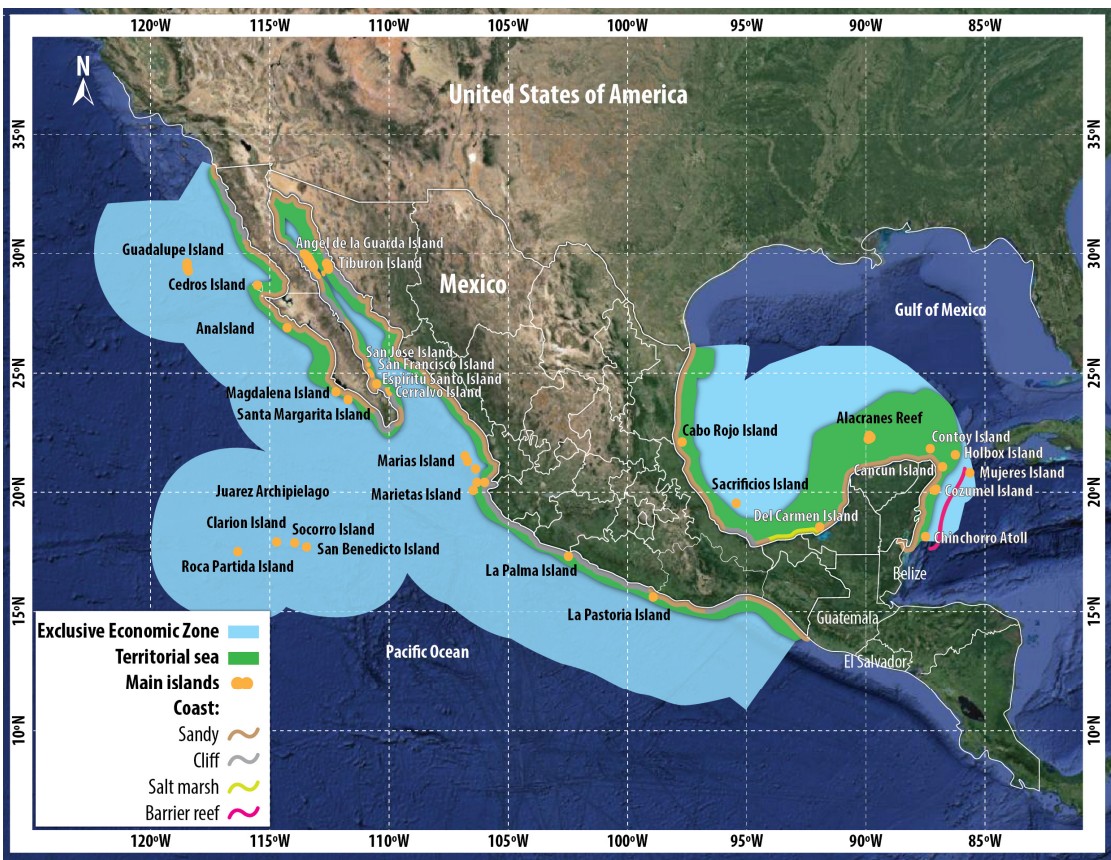

**Figure 2.** Continental shelf, territorial sea, and exclusive economic zone of Mexico with the main island areas.

Both agriculture and aquaculture exert strong pressure on water and soil resources and may result in negative environmental impacts if not properly managed. The global agri-food sector is being forced to produce an increasing volume of food to an estimated population of 9.7 billion people in 2050. This is only part of the challenge, as it also seeks to achieve it with minimal impact on ecosystems and in the context of global climate change. In global terms, the technological development of aquaculture is very advanced to avoid or reduce impacts that contribute to the degradation of ecosystems. However, technology transfer to countries such as Mexico faces challenges that hinder implementation; for example, there is a lack of capabilities (professionalization of the activity) and financial investment, and limited access for most producers to cutting-edge technology due to costs or technical capabilities [24]. Food production through aquaculture can exacerbate the effects of climate change in the country by fostering the degradation of habitats, the modification of coastlines, changes in land use, wastewater discharges, the use of fossil fuels, and conflicts among users competing for space and resources. This scenario must be avoided through strategic investment in planning, carrying capacity studies, capacity development, and professionalization. In a transversal way, investment in science and technology development is essential to innovate and adapt cutting-edge technologies to the national context, which gives rise to designing alternative production systems, such as aquaponics [51].

Although aquaculture and hydroponics have been practiced since ancient times, and in Mexico there are important archaeological vestiges of it, the combination of the two elements in aquaponics is relatively recent [52,53]. The first reports in this field date back to the 1970s, and until the 1980s, these developments had limited application [54]. It was not until the mid-1990s that the first aquaponic system was developed with tilapia (*Oreochromis niloticus*) and tomato [55]. From then on, tests and ventures of experimental

systems and commercial aquaponic farms began and have had great success in Baja California, Jalisco, Sonora, and Veracruz [48,56–61]. According to this panorama, important experience has been gained with these technologies that may trigger the development of ColdAg in the country as part of a wider application of the OTEC technology.

Priority and more suitable areas for the implementation of the OTEC system in Mexico have been determined [36] (Figure 3), most of them on the Pacific Ocean slope. All the areas meet essential criteria: (1) the maximum distance between the cold deep water pumping point to the coastline is less than 10 km, taking the 1000 m depth isobath as a reference; (2) a temperature difference or a thermal gradient of 22 to 26 °C, which allows the system to operate 24 h a day; (3) the distance at some points of interest to the electrical distribution node is close to 14 km, which is the maximum distance to the closest electrical substation at any of the sites of interest [62]; and (4) the existence of communities whose electricity consumption is less than or equal to 212 MW a year [63,64]. Importantly, 13.28% of the homes in the coastal zone of the Pacific Ocean do not have electricity [65]. Hotel occupancy must also be considered since many of the priority areas are places with great tourist acceptance. In addition to homes, hotels and other service centers can also benefit from the seawater air conditioning system and the production of drinking water. An integrative exploitation of the OTEC technology, including not only the generation of electrical energy but also seawater air conditioning, desalination, chilled-soil agriculture, and aquaculture, would be a feasible strategy to improve the overall economy of the whole system and in turn contribute to the sustainable development of the communities.

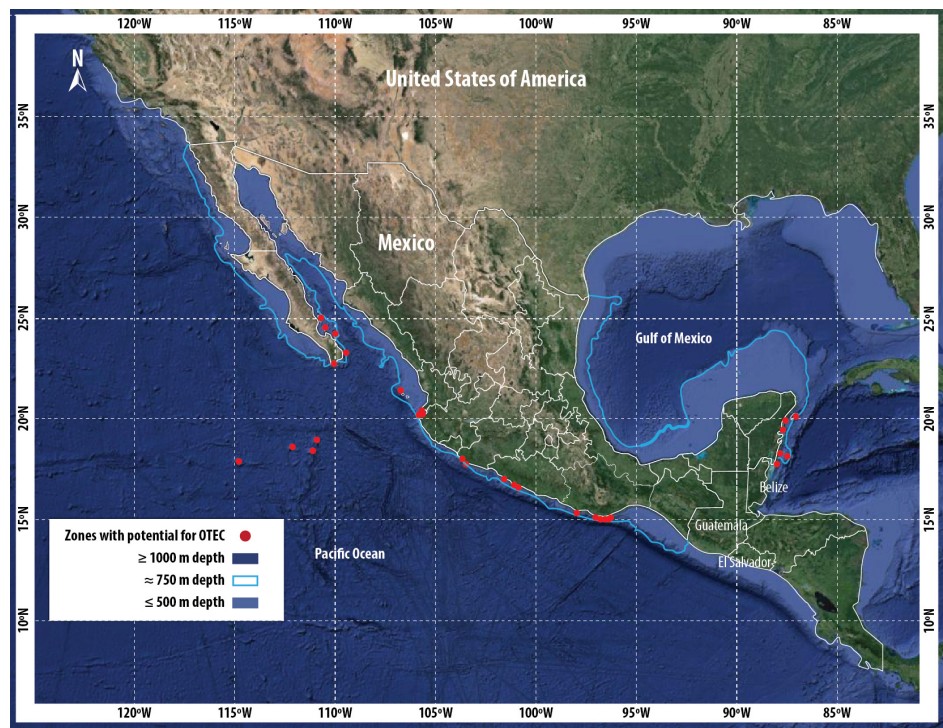

**Figure 3.** Potential sites for the implementation of the Ocean Thermal Energy Conversion (OTEC) system in Mexico.

Among the important sites fronting the Mexican Pacific Ocean that meet the four aforementioned criteria are the following: Bahía Tangolunda, San Agustín Huatulco, Puerto Ángel, El Coyote, Agua Blanca, Los Naranjos, and El Azufre in Oaxaca; Nuxco, San Luis de la Loma, El Cobano, and Barra de Potosí in Guerrero; Colola and Ostula in Michoacán; Bahía de Banderas in Jalisco; and Los Frailes, Cabo Pulmo, and Cabo San Lucas in Baja California Sur [65]. These sites present significant food and economic lags and do not have industrial or maritime port infrastructure. However, with technological implementation,

these facilities could be used in the future, with the distance to the coastline with reference to the point of obtaining cold water ranging between 2 and 10 km [36].

As for the islands on the Pacific Ocean side, Cerralvo Island, the Revillagigedo Archipelago, the Marías Islands, and the Marietas Islands could be considered as possible sites for the installation of the OTEC system, especially considering that they are small sites, where food through aquaculture and agriculture, drinking water and air conditioning could be provided [65]. On average, distances from the coastline to the cold-water extraction zone range from 1 to 3 km. Nevertheless, the thermal gradient decreases considerably during the autumn and winter months, so the plant could only operate for a maximum of six months, and they also have the disadvantage that they are protected natural areas, with the exception that, in the case of Marías Islands, the area became a natural museum, so it could be of great interest [46].

In the case of the Gulf of Mexico and the eastern part of the Gulf of Tehuantepec, the great drawback for access to this technology is that although they have regions where the temperature difference is greater than 20 °C, the continental shelf extends up to 100 km in distance, so the cold-water extraction areas are too far away, generating high costs for the placement of pipes, as well as for their extension [66]. However, in the region of San Andrés Tuxtla, Veracruz, there is an area where this technology can be exploited, through the closed cycle, 15 km away from the coastline, generating up to 109 MW of electricity [67]. In the case of the north of the Baja California Peninsula and the northwest of the Gulf of California, the drawback is that the thermal gradient is so low that the OTEC plant would only operate for three or four months in the summer. This is because the presence of the California Current generates a very considerable cooling of the surface.

The physical characteristics of the Mexican Caribbean Sea in the exclusive economic zone (EEZ) confer broad possibilities for commercial exploitation in a thermal gradient that has an estimated gross potential of around 30,000 MW at depths greater than 800 m and less than 1000 m [68]. The potential places to build an OTEC plant at distances less than 10 km from the coast are as follows: Cozumel Island, Punta Allen, Tulum, Sian Ka'an, Xcalac, Mahahual, and Banco Chinchorro in the state of Quintana Roo. For Cozumel Island, the 700 m deep isobath is located at 4 km and 6.7 km to the 1000 m line on the southeast coast and from the coast to the city of Cozumel 30 km. Punta Allen is a town that lacks electricity, drinking water, and sewage services, located 50 km from Tulum. The OTEC point of interest is located 40 km from the coast to Tulum. The 700 m and 800 m isobaths are the closest to the 10 km limit. Sian Ka'an is a Biosphere Reserve, so installing a plant could be complicated, although this type of plant would help desalinate the water for human consumption. Xcalac, Mahahual, and Banco Chinchorro represent a large area with great potential to install an OTEC plant, from Xcalac to Mahahual, with a distance of 6.7 to 14.4 km. The closest city is Chetumal, which is located at 55 km [66].

Thus, summarizing, the priority areas should be considered from Nayarit to Oaxaca in the Pacific Ocean and the western part of the Yucatán Channel in the Caribbean Sea [36].

Mexico has the necessary oceanographic conditions for the construction of the OTEC system, and the potential use of cold-water recirculation in soil cooling systems for the production of crops from temperate climates under tropical climates. The number of benefits that this technology offers is evident, and the national research centers have a lot to contribute to make it accessible and available at low cost. Although for now the maximum energy potential obtained by the cycle is 1 MW (Republic of Kiribati, French Polynesia), a plant can offer energy autonomy to countries that today depend on oil and that the construction of this type of plant promotes the local economy based on new potential markets that are currently known as the "Industry of the Depths" [36], previously described (i.e., energy, air conditioning, alt, drinking water, aquaculture and agriculture, among others).

The costs of OTEC plants can be estimated from different parameters [66–73]:

- Electrical capacity and classification: Plants with greater electrical capacity and built on platforms could have lower costs (USD 2650 USD $kW^{-1}$) than those with lower capacity (1 to 10 MW) built on floating systems (USD 16,400 to 35,400 USD $kW^{-1}$).
- Type of cycle: The closed cycle design is slightly more economical than the open cycle.
- Byproducts: Fresh water would increase the installation cost but would improve the economy in regions where there is high demand.
- Distance from the thermal source to the coast: The greater the distance between the coastline (greater than 10 km) and the source of cold water, the higher the cost and energy loss.
- Thermal potential: The better the marine thermal potential where the plant is intended to be built, the greater its thermal, energy, and economic efficiency.
- Availability: To avoid economic losses, it must be ensured that the plant operates most of the time.

Other parameters to consider are as follows: local infrastructure, possible ecological impact, climatic agents, and physical environment, which could affect the continued operation of the plant [66]. Operation and maintenance costs depend on the classification of the plant: (a) onshore, at least 10 workers would be needed and (b) floating systems (offshore), 17 workers would be needed. In addition, it must be considered that 12 operators are needed to work all year round (24 h/365 days) and at least five engineers or people in charge of plant administration [74]. The production cost depends on the life of the plant and a 3% constant inflation must be assumed, knowing that the larger the size of the plant, the lower the cost of electricity production [66].

To prove that the system is profitable and competitive, it is necessary to carry out an economic analysis prior to its implementation, which includes detailed analyses of capital costs expressed (USD $kW^{-1}$), extraction costs of fuel, sociocultural costs, corrosion costs, land use, and subsidies. OTEC plants represent a large capital investment as they also demand operational and design expertise. However, the economy of the scale makes it more financially viable (investment in large plants). For the development of this technology in Mexico, it is advisable to minimize investment costs, obtain the highest percentage of financing possible, and achieve the commercialization of the byproducts to positively impact the financial indicators and promote this type of technology [75].

In Mexico, both the OTEC system and ColdAg technology can be implemented in the medium and long terms in different sites exhibiting optimal thermal gradient and relative humidity [40,66,76,77]. Appropriate values of relative humidity and temperature are found in most of the country's coastal regions [78]. This is very important since the ColdAg principle implies that environmental humidity must be condensed by the cold temperature of deep seawater, for the water to irrigate the crops and for the nutrients to access the root for potential absorption [40]. In terms of plant cell signal perception and transduction, when the soil temperature is cooler than the leaf temperature, the plant receives the signal and responds by growing as if it was spring [79].

## 4. Applications and Exploitation of the Thermal Gradients of the Oceans

For now, the cold water that comes from the depth of the sea has been used for the agriculture of temperate climate crops in places with tropical climates. At such sites, cold water is pumped through a series of hoses that are embedded in the soil to cool it and ensure the production of crops in hydroponics [80].

In Goseong, South Korea, next to the OTEC and seawater utilization plant, high-technology industries producing mineral salts, mineral waters, and cosmetics have been established. Waters linked to OTEC operation are being used to produce crops in hydroponic systems, for shellfish and fish farms, and for refrigeration plants providing cold water and air conditioning, all of them contributing to sustainable development goals. In the crop hydroponic systems, deep seawater is diluted by 70% so that the plants can take advantage of the minerals from the seabed and, at the same time, use the diluted seawater for irrigation [81].

In the United States, cold seawater is being used to irrigate grapevines and produce different vegetables and ornamental species [79].

In Malaysia, the use of soil cooling is useful for the growth of crops from temperate climates under tropical climates using lettuce as a plant model [81]. When the plants were exposed to cold soil, they showed greater average weight and development of their roots, stems, and leaves in the three growth cycles, while the microbiota benefited from such conditions [82]. The limitation of these studies was that the soil cooling system they used could only produce small-scale crops.

With the ColdAg system, it is possible to produce temperate climate crops in tropical climate conditions all year round. Coupled with the use of the recirculation of cold deep seawater, derived from the OTEC process and combined with organic fertilizers and other agricultural practices, commercial crops could be produced on a large scale, which could be an answer to the problem of food poverty in coastal regions. It is estimated that 23.5% of Mexico's population lives in food poverty [18] and 18.9% of children suffer from chronic malnutrition [83], although this number dropped to 13.9% as economic poverty decreased by three percentage points in 2022 [18]. Despite this, this issue remains an entrenched problem that needs to be addressed decisively.

Chronic malnutrition affects 7.7% of the country's urban inhabitants and 11.2% of its rural residents. Of the total malnourished population in the rural areas, 55% occurs in the coastal regions of the country. This panorama, combined with the effects of climate change on the availability of fresh and safe food, as well as access to energy and drinking water, makes it necessary for Mexico's islands and coastal regions to know the optimal management of crop production systems. Fortunately, the oceans surrounding islands or fronting coastal regions may offer solutions, since a new economic base could help these regions meet their basic needs through a circular and sustainable industry or economic sector [84]. The use of atmospheric humidity is an opportunity to mitigate the problem of scarce good-quality water. It is therefore recommended to carry out tests of its implementation in the production of local crops in areas where there is limited water availability to improve living conditions and promote welfare.

However, the economic benefits of OTEC remain extremely low. These technologies still face various research and development challenges if they are to become viable and competitive alternatives and allow governments to make regulations for their implementation. Integrative approaches that include seawater utilization systems for energy, water, and food would be a feasible strategy to improve the overall economy [85,86].

## 5. Conclusions

OTEC technology remains in constant development and expansion. While in Mexico it may be in its infancy, it has the potential to be incorporated into the global energy matrix in the medium and long term. Mexico and other countries with important coastal regions could develop their own technologies to have OTEC plants in operation. In addition to energy, these OTEC plants may provide facilities to produce drinking water, air conditioning, and food (i.e., agriculture and aquaculture), among other benefits. In the case of cold-water agriculture (ColdAg), there is little current scientifically supported information. Therefore, there is a dire need to develop, test and adapt novel technologies and innovations aimed at taking advantage of this OTEC variant and contributing to the provision of goods and services for coastal regions and islands in a sustainable environment. Consequently, this strategy may substantially contribute to meeting the 17 Goals for Sustainable Development, especially those related to the fight against poverty, hunger, and inequity, while contributing to good health and well-being, clean water, and sanitization [87]. Furthermore, this technology may speed the development of the so-called "industry of the depths", which may provide clean and affordable water, energy, minerals, and food in a cost-effective, environmentally friendly manner. High-technology industries aimed at the production of mineral salts, mineral waters, and cosmetics may result from the sustainable exploitation of this technology.

Mexico is a good candidate to profit from the OTEC system and ColdAg, since there are climatic and oceanographic conditions that may allow the use of products derived from the implementation of this technology. In the coastal regions of the country there are some areas with adequate conditions to adopt the ColdAg technology, especially on the slopes of the Gulf of Mexico in the southern part of San Andrés Tuxtla, Veracruz, the area of the Caribbean Sea in Quintana Roo, and the slope of the Mexican Pacific Ocean from the coast of Nayarit to Oaxaca. In these areas, the climate presents temperature values greater than 30 °C, which guarantees a thermal differential greater than 20 °C if the coastal soil cools by 3 or 4 °C. This could allow to produce certain temperate crops under protected agriculture or open field conditions, providing access to better fresh and nutritious foods, and probably lowering production costs. In the case of Mexico, research on this topic will be crucial and fundamental to exploit these technologies to contribute to the fight against food poverty and inequity, thus supporting the national objective of achieving food security and sovereignty. Nevertheless, there remains a wide gap between the need for large ColdAg systems and current technical capabilities.

**Author Contributions:** Conceptualization, A.G.-H. and F.C.G.-M.; methodology, A.G.-H.; validation, L.I.T.-T. and A.L.-H.; formal analysis, A.G.-H. and A.G.-H.; investigation, A.G.-H., A.L.-H. and L.I.T.-T.; resources, A.G.-H. and L.I.T.-T.; writing—original draft preparation, A.G.-H.; writing—review and editing, F.C.G.-M.; supervision, F.C.G.-M. and L.I.T.-T.; project administration, F.C.G.-M.; funding acquisition, F.C.G.-M. and L.I.T.-T. All authors have read and agreed to the published version of the manuscript.

**Funding:** This research was funded by the National Council of Humanities, Sciences and Technologies (CONAHCYT) of Mexico, grant number 562620 of Postdoctoral Residences for Mexico. The APC was funded by the College of Postgraduates in Agricultural Sciences. The funders had no role in the design, execution, interpretation, or writing of the study.

**Data Availability Statement:** The data presented in this study are openly available in the list of references at the end of this review.

**Acknowledgments:** To the Mexican Center for Innovation in Ocean Energy (CEMIE-O) for its collaboration in the search for bibliographic references and to Tom Daniel, Luis A. Vega, Yasuyuki Ikegami and Hyeon Ju Kim for their contribution of relevant data to the preparation of this manuscript.

**Conflicts of Interest:** The authors declare no conflicts of interest.

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
