# Peer review of "Potential of Cold-Water Agriculture (ColdAg) in Mexico: Challenges and Opportunities for Sustainable Food Production"

_sustainability, doi:10.3390/su16104298_

Round 1
Reviewer 1 Report
Comments and Suggestions for Authors
The manuscript “Cold Water Agriculture (ColdAg): Challenges and Opportunities for Sustainable Food Production in the Future” reports a literature study on Cold Water Agriculture concept.
This is quite an old concept, with more than three decades hystory. The claim of new technology is not supported. Eventually, the authors should be more specific in this regard.
The abstract section is not the adequate place to explain a concept, but to explain shortly the topic, goals and main conclusions of the manuscript.
Line 27, statement is incorrect, relative humidity is a measure of the water content of air. Relative humidity cannot condens.
Line 128,”4.2 million Mg of agricultural pollutants”? Use International System of Units
Lines135-140: the authors reitterate the definition of ColdAg concept, stating again (like in the abstract section) the objective of the concept, and forget to present the actual objective of the manuscript. The structure of the manuscript must be explained in this paragraph also, and the corresponding choices of the authors as their view to meet the general purpose of the study. For instance, the need for a distinct fourth section is not clear.
Paragraph at lines 222-226 contains multiple errors.
The conclusions section highlights the advantages of OTEC. Yet, it should stress the findings of the study.
The reference section is outdated. The authors should study and include also literature studies published in 2023.
Author Response
Reviewer 1
The manuscript “Cold Water Agriculture (ColdAg): Challenges and Opportunities for Sustainable Food Production in the Future” reports a literature study on Cold Water Agriculture concept.
This is quite an old concept, with more than three decades history. The claim of new technology is not supported. Eventually, the authors should be more specific in this regard.
Agreed. We have reviewed and updated this information in the revised version.
The abstract section is not the adequate place to explain a concept, but to explain shortly the topic, goals and main conclusions of the manuscript.
Agreed. We have reviewed and re-edited this information in the revised version.
Line 27, statement is incorrect, relative humidity is a measure of the water content of air. Relative humidity cannot condense.
Agreed. We have reviewed and corrected this information in the revised version.
Line 128,”4.2 million Mg of agricultural pollutants”? Use International System of Units
The ton (t) is a unit of mass. Even though it is a metric unit, it is not a proper SI unit. The megagram is the proper SI unit with equal value to the ton. One megagram is equal to 1000 kilograms.
Lines135-140: the authors reiterate the definition of ColdAg concept, stating again (like in the abstract section) the objective of the concept, and forget to present the actual objective of the manuscript.
Agreed. We have reviewed and updated this information in the revised version.
The structure of the manuscript must be explained in this paragraph also, and the corresponding choices of the authors as their view to meet the general purpose of the study. For instance, the need for a distinct fourth section is not clear.
Agreed. We have reviewed and re-edited this information in the revised version. The content of the paragraphs corresponding to the forms section 4 was included as part of the section 3.
Paragraph at lines 222-226 contains multiple errors.
We have reviewed and corrected this information in the revised version.
The conclusions section highlights the advantages of OTEC. Yet, it should stress the findings of the study.
Agreed. We have reviewed and corrected this information in the revised version.
The reference section is outdated. The authors should study and include also literature studies published in 2023.
In this review we aimed to collect both classic and current literature on the topic. In the revised version we have consider more references recently published.
Reviewer 2 Report
Comments and Suggestions for Authors
In-depth analysis in the manuscript is missing. The introduction does not cover the title, also misses an important aspect of the manuscript. the discussion section is tool week, i would not recommend this manuscript for the publication in this journal, The manuscript is poorly presented and does not fulfill the criteria and scope of this journal.
Comments on the Quality of English LanguageIn-depth analysis in the manuscript is missing. The introduction does not cover the title, also misses an important aspect of the manuscript. the discussion section is tool week, i would not recommend this manuscript for publication in this journal, The manuscript is poorly presented and does not fulfill the criteria and scope of this journal.
Author Response
Reviewer 2
In-depth analysis in the manuscript is missing.
We have included new references and updated the analysis making it deeper
The introduction does not cover the title, also misses an important aspect of the manuscript.
We have reviewed and corrected this information in the revised version.
The discussion section is tool week, I would not recommend this manuscript for the publication in this journal, The manuscript is poorly presented and does not fulfill the criteria and scope of this journal.
We have substantially improved the manuscript, taking into consideration all comments and concerns made by the reviewers.
Reviewer 3 Report
Comments and Suggestions for Authors
The manuscript titled "Cold Water Agriculture (ColdAg): Challenges and Opportunities for Sustainable Food Production in the Future" reviewed the importance, development, background and application cases of the ColdAg technology, and explored the application potential in Mexico. Some suggestions I proposed are:
1. "ColdAg" or "Cold water agriculture" is appropriate for this technology? This technology is based on the OTEC technology. No papers were checked in the WOS system use the keyword "ColdAg".
2. The authors are telling a very long and specifically story by detailed give the theory of the OTEC technology. Most of the paragraphs can be shorten.
3. Some data, especially the number need to give the source.
Author Response
Reviewer 3
The manuscript titled "Cold Water Agriculture (ColdAg): Challenges and Opportunities for Sustainable Food Production in the Future" reviewed the importance, development, background and application cases of the ColdAg technology, and explored the application potential in Mexico. Some suggestions I proposed are:
- "ColdAg" or "Cold water agriculture" is appropriate for this technology? This technology is based on the OTEC technology. No papers were checked in the WOS system use the keyword "ColdAg".
The term “ColdAg” is correct.
Some references recently published in high impact journals or publishers do include this term:
Kim AS, Kim HJ. 2020. Ocean Thermal Energy Conversion (OTEC). Past, Present, and Progress. Intech Open: London, UK. http://dx.doi.org/10.5772/intechopen.86591.
Garduño-Ruiz, E.P.; Silva, R.; Rodríguez-Cueto, Y.; García-Huante, A.; Olmedo-González, J.; Martínez, M.L.; Wojtarowski, A.; Martell-Dubois, R.; Cerdeira-Estrada, S. Criteria for Optimal Site Selection for Ocean Thermal Energy Conversion (OTEC) Plants in Mexico. Energies 2021, 14, 2121. https://doi.org/10.3390/en14082121
Tobal-Cupul, J.G.; Garduño-Ruiz, E.P.; Gorr-Pozzi, E.; Olmedo-González, J.; Martínez, E.D.; Rosales, A.; Navarro-Moreno, D.D.; Benítez-Gallardo, J.E.; García-Vega, F.; Wang, M.; et al. An Assessment of the Financial Feasibility of an OTEC Ecopark: A Case Study at Cozumel Island. Sustainability 2022, 14, 4654. https://doi.org/10.3390/su14084654
This concept is also referred as “refrigerated soil agriculture” or “chilled-soil agriculture” in other references:
Herrera, J.; Sierra, S.; Ibeas, A. Ocean Thermal Energy Conversion and Other Uses of Deep Sea Water: A Review. Journal of Marine Science and Engineering 2021, 9, 356. https://doi.org/10.3390/jmse9040356
Hu, Z., Deng Z., Gao, W., Chen, Y. (2023). Experimental study of the absorption refrigeration using ocean thermal energy and its under-lying prospects. Renewable Energy 213, 2023: 47-62. https://doi.org/10.1016/j.renene.2023.05.086
- The authors are telling a very long and specifically story by detailed give the theory of the OTEC technology. Most of the paragraphs can be shorten.
Agreed. Some of the paragraphs have been re-edited and shorten.
- Some data, especially the number need to give the source.
This issue was fixed in the revised version.
Reviewer 4 Report
Comments and Suggestions for Authors
The article’s significance of content is high. This study attempts to provide an overview of the opportunities and challenges of cold-water agriculture in future food production. However, the background of cold-water agriculture and related reference materials are not sufficiently provided in this paper. Specific suggestions are as follows:
1. In the Introduction, please clarify more distinctly the connection between global agricultural challenges and the potential of ColdAg technology. Specifically, elaborate on how ColdAg directly addresses these challenges in greater detail.
2. A brief overview of other contemporary solutions to the challenges posed by climate change in agriculture, and how ColdAg complements or surpasses these solutions, will provide a more comprehensive background.
3. In section 2, "The concept of Cold Water Agriculture (ColdAg)," the definition of ColdAg is not clearly and accurately stated at the beginning.
4. In section 2, "The concept of Cold Water Agriculture (ColdAg)," please include potential limitations or challenges of ColdAg .
5. In section 3. Possibilities for implementing ColdAg in Mexico, please add the similarities between Korea and Mexico so as to ensure the successful implementation of ColdAg in Mexico.
6. Although the manuscript indicates that ColdAg could solve the problem of food poverty in coastal areas, it lacks a detailed analysis of the economic viability and sustainability of expanding ColdAg systems (Lines 551-571).
7. The same as point 6, indicating repetition and emphasizing the need for a detailed analysis on economic viability and sustainability of ColdAg systems expansion (Lines 551-571).
Comments on the Quality of English Language
Minor editing of English language required
Author Response
Reviewer 4
The article’s significance of content is high. This study attempts to provide an overview of the opportunities and challenges of cold-water agriculture in future food production. However, the background of cold-water agriculture and related reference materials are not sufficiently provided in this paper. Specific suggestions are as follows:
- In the Introduction, please clarify more distinctly the connection between global agricultural challenges and the potential of ColdAg technology. Specifically, elaborate on how ColdAg directly addresses these challenges in greater detail.
Agreed. This issue was addressed in the revised version. We have given a more broad context since the problems imposed by climate change impacts, poverty an malnutrition have given rise to serious challenges in our country.
- A brief overview of other contemporary solutions to the challenges posed by climate change in agriculture, and how ColdAg complements or surpasses these solutions, will provide a more comprehensive background.
Agreed. This issue was addressed in the revised version.
- In section 2, "The concept of Cold Water Agriculture (ColdAg)," the definition of ColdAg is not clearly and accurately stated at the beginning.
Agreed. This issue was fixed in the revised version.
- In section 2, "The concept of Cold Water Agriculture (ColdAg)," please include potential limitations or challenges of ColdAg.
Agreed. The potential limitations of the technology were also addressed in the revised version.
- In section 3. Possibilities for implementing ColdAg in Mexico, please add the similarities between Korea and Mexico so as to ensure the successful implementation of ColdAg in Mexico.
Agreed. This information was included in the revised version. The paragraphs related have been highlighted with yellow shadows and letter in blue color.
- Although the manuscript indicates that ColdAg could solve the problem of food poverty in coastal areas, it lacks a detailed analysis of the economic viability and sustainability of expanding ColdAg systems (Lines 551-571).
We have included the following paragraph in the revised version:
However, the economic benefits of OTEC remain extremely low. These technologies still face various research and development challenges if they are to become viable and competitive alternatives and allow governments to make regulations for their implementation. Integrative approaches that include sea water utilization systems for energy, water, and food would be a feasible strategy to improve the overall economy.
- The same as point 6, indicating repetition and emphasizing the need for a detailed analysis on economic viability and sustainability of ColdAg systems expansion (Lines 551-571).
We have included the paragraph cited in the comment above.
Reviewer 5 Report
Comments and Suggestions for Authors
Dear Authors,
I read the article with interest, given its topic. Still, I consider that some issues need to be addressed.
1. The novelty of the research must be emphasized.
2. The article is too 'diluted,' meaning that much information is irrelevant to the study. At least a fourth of the article must be removed to provide concise and relevant information. As it is, the paper is more a 'general communication' rather than a scientific paper.
3. The article lacks clear organization. A chart flow might help the reader understand the direction of the research. Information must be better grouped into different sub-sections.
Given the above, the article must be modified before being considered for publication.
Comments on the Quality of English LanguageMinor corrections must be made.
Author Response
Reviewer 5
- The novelty of the research must be emphasized.
Agreed. We have reviewed and re-edited the whole manuscript, highlighting different sections that required our attention.
- The article is too 'diluted,' meaning that much information is irrelevant to the study. At least a fourth of the article must be removed to provide concise and relevant information. As it is, the paper is more a 'general communication' rather than a scientific paper.
Agreed. The manuscript was re-edited and shortened.
- The article lacks clear organization. A chart flow might help the reader understand the direction of the research. Information must be better grouped into different sub-sections.
The manuscript was reorganized, and some sections were shortened. We believe that the manuscript is now more comprehensive.
Round 2
Reviewer 1 Report
Comments and Suggestions for Authors
The manuscript “Cold Water Agriculture (ColdAg): Challenges and Opportunities for Sustainable Food Production in the Future” has been revised and significantly improved compared to the original submitted form.
The authors addressed my comments and recommendations.
However, the manuscript still presents several minor issues.
(1) Line 28: remove the article from “the some”.
(2) Line 29: “the feasibility of Mexico” is wrongly used. The authors should reformulate to stress the feasibility of the technology.
(3) Lines 232-233, “and the peace” should be removed.
Author Response
Reviewer 1
The manuscript “Cold-Water Agriculture (ColdAg): Challenges and Opportunities for Sustainable Food Production in the Future” has been revised and significantly improved compared to the original submitted form.
The authors addressed my comments and recommendations.
However, the manuscript still presents several minor issues.
(1) Line 28: remove the article from “the some”.
Agreed. The article “the” was removed from the revised version.
(2) Line 29: “the feasibility of Mexico” is wrongly used. The authors should reformulate to stress the feasibility of the technology.
Agreed. The corresponding correction was done in the revised version.
(3) Lines 232-233, “and the peace” should be removed.
Agreed. The corresponding correction was done in the revised version.
Reviewer 4 Report
Comments and Suggestions for Authors
The author attaches great importance to the opinions of the reviewers and has made good revisions.
Author Response
Reviewer 2
The author attaches great importance to the opinions of the reviewers and has made good revisions.
Thank you for your suggestions and recommendations that allowed us to improve our review.